# A Novel Algorithm for Feature Selection Using Penalized Regression with Applications to Single-Cell RNA Sequencing Data [note 2]

**DOI:** 10.3390/biology11101495

**Published:** 2022-10-12

**Authors:** Bhavithry Sen Puliparambil, Jabed H. Tomal, Yan Yan

**Affiliations:** 1Master of Science in Data Science Program, Thompson Rivers University, 805 TRU Way, Kamloops, BC V2C 0C8, Canada; 2Department of Mathematics and Statistics, Thompson Rivers University, 805 TRU Way, Kamloops, BC V2C 0C8, Canada; 3Department of Computing Science, Thompson Rivers University, 805 TRU Way, Kamloops, BC V2C 0C8, Canada

**Keywords:** high-dimensional data, single-cell RNA sequencing, gene expression data, machine learning, penalized regression, lasso, sparse group lasso, feature selection, R

## Abstract

**Simple Summary:**

Single Cell RNA Sequencing generates gene expression data at a single cell resolution. While single cell RNA has many applications in biomedical research, the high dimensionality of the data produced poses a considerable computational challenge. This study proposes a novel algorithm using penalized regression methods to analyze single cell RNA sequencing data. The proposed algorithm reduces high dimensionality of the gene expression data using a sequence feature selection methods such as Ridge regression, LASSO, Elastic Net, Drop LASSO, and Sparse Group LASSO. The proposed algorithm successfully detected highly differentiated genes, including the marker genes, for 5 different single cell RNA sequencing datasets associated with the species mouse, plant, and human.

**Abstract:**

With the emergence of single-cell RNA sequencing (scRNA-seq) technology, scientists are able to examine gene expression at single-cell resolution. Analysis of scRNA-seq data has its own challenges, which stem from its high dimensionality. The method of machine learning comes with the potential of gene (feature) selection from the high-dimensional scRNA-seq data. Even though there exist multiple machine learning methods that appear to be suitable for feature selection, such as penalized regression, there is no rigorous comparison of their performances across data sets, where each poses its own challenges. Therefore, in this paper, we analyzed and compared multiple penalized regression methods for scRNA-seq data. Given the scRNA-seq data sets we analyzed, the results show that sparse group lasso (SGL) outperforms the other six methods (ridge, lasso, elastic net, drop lasso, group lasso, and big lasso) using the metrics area under the receiver operating curve (AUC) and computation time. Building on these findings, we proposed a new algorithm for feature selection using penalized regression methods. The proposed algorithm works by selecting a small subset of genes and applying SGL to select the differentially expressed genes in scRNA-seq data. By using hierarchical clustering to group genes, the proposed method bypasses the need for domain-specific knowledge for gene grouping information. In addition, the proposed algorithm provided consistently better AUC for the data sets used.

## 1. Introduction

Single-cell RNA sequencing (scRNA-seq) technology is very popular in biomedical research. This technology examines the gene expressions among different cells in a tissue and provides the gene expression profile on a cell level [1]. Compared with the traditional RNA-seq technology, which examines the average gene expressions in a tissue, scRNA-seq is a recent development and a very advanced technology.

scRNA-seq technology is also useful for identifying cellular heterogeneity. On the other hand, scRNA-seq has low capture efficiency and high dropouts compared with regular RNA-seq technologies. Yet, quality control is necessary for removing technical noise from scRNA-seq data [2]. scRNA-seq protocols can be classified into two categories, full-length transcript sequencing approaches and 3${}^{\prime }$-end or 5${}^{\prime }$-end transcript sequencing [2]. In addition, 3${}^{\prime }$-end or 5${}^{\prime }$-end transcript sequencing are known as unique molecular identifier (UMI) tag-based protocols, and full-length transcript sequencing is known as a non-UMI-based protocol [3]. UMI tag-based scRNA-seq protocols uses UMI tags for different transcript molecules. During the scRNA-seq process, the transcript molecules get attached with their respective UMI tags, and then these UMIs are counted to obtain the number of transcripts for each gene [4]. UMI tag-based protocols include Drop-seq and 10× Genomics Chromium. Compared with UMI tag-based sequencing, the Non-UMI-based scRNA-seq protocols sequence whole transcripts [2]. Non-UMI-based protocols include Smart-seq2, MATQ-seq, and Fluidigm C1 [3].

One of the many applications of scRNA-seq technology is differentiating cell groups by comparing their molecular signatures, for instance, identifying highly differentiated genes to cluster knockout and wild type cells. Similarly, by comparing the gene expression profile of cancer cells with that of healthy cells, one can identify genes with altered expression that might be responsible for cancer. The applications of scRNA-seq technology are numerous and of high impact in genomic research. However, the scRNA-seq data come with certain challenges as well [5], especially requiring high computation time and resources. Methods dealing with these challenges typically include shifting and scaling, batch effect correction, dimensionality reduction, missing data imputation, and selection of important features. High dimensionality is a major computational challenge in analyzing scRNA-seq data. Therefore, dimensionality reduction methods such as principal component analysis (PCA), T-distributed stochastic neighbor embedding (t-SNE), and feature selection are often performed in scRNA-seq data analysis [2].

scRNA-seq data usually contain a larger number of features (genes) than the number of samples (cells). People typically represent *p* as the dimension of features and *n* as the dimension of samples. When p>>n, we call the data having the “large-p-small-n” problem. Statistical models could result in poor prediction performance due to over-fitting when training data contain fewer samples compared to the number of features [6]. There are several methods to deal with the “large-p-small-n” problem in Machine Learning (ML), of which feature selection is the most useful.

Feature selection is the method of selecting variables that contain better signal than the noise variables for the target variable of interest Random forests, Recursive Feature Elimination (RFE), and penalized regression are often used for feature selection in ML.

In this research, we explored the penalized regression methods to select feature variables over RFE and random forest. The rationale behind this choice is that RFE is computationally expensive when applied to high-dimensional data [7]. Therefore, RFE is not an ideal choice for feature selection in scRNA-seq data. Random forest and its application on scRNA-seq data have been examined thoroughly in other studies [8,9,10,11]. Moreover, some penalized regression methods are worth exploring to compare their performances for feature selection in scRNA-seq data because these methods were developed primarily for tackling the challenges of “large-p-small-n” problems [12].

Some popular variants of penalized regression in machine learning are ridge regression, least absolute shrinkage and selection operator (lasso) [13], and elastic net regression [14], where the latter is a combination of ridge and lasso. The applicability of these methods depends on the problem one is addressing in modeling scRNA-seq data. Ridge regression reduces the dimension of feature variables by making their coefficients closer to 0. The features for which the coefficients are close to zero are forced to be excluded while the remaining features are retained in the model. This notion of feature selection using ridge regression is known as hard thresholding. Lasso regression [13] implements soft thresholding for feature selection. In soft thresholding, the regression coefficients are made exactly equal to zero. The limitation of penalized regression such as ridge, lasso, and elastic net regression is that these algorithms do not consider any previously known relationships between the features, such as the grouping information of the genes in the scRNA-seq data, while selecting them [15,16]. There are improved lasso methods such as group lasso [17] and sparse group lasso [18] that allow us to use grouping information of features, which could account for the interrelationship of variables. Drop lasso [12] and big lasso [19] are two other interesting variants of penalized regression. Our primary objective in this article is to explore scRNA-seq data specific to different species using the collection of penalized regression methods, and determine which method is more suitable for the problem under study.

There are some other variants of lasso regression such as nuisance penalized regression [20], fused lasso [21], adaptive lasso [22], and prior lasso [23]. Nuisance penalized regression is suitable when some of the feature variables are of particular interest and others are considered noise (nuisance). However, this distinction between genes is not always known in advance. In this study, we are comparing methods that do not require prior knowledge of genes for feature selection. Therefore, nuisance penalized regression is not selected for analysis in our research. While prior lasso can be applied to biological data, it requires prior information to be incorporated [23]. Fused lasso was proposed for processing image-based time series data, which is very different from scRNA-seq application. Meanwhile, adaptive lasso incorporates penalty to proportional hazards regression. These methods may not be suitable for our applications. Therefore, they are excluded from our study.

There exist studies in penalized regression methods for high-dimensional data [24,25]. However, these studies did not include a performance comparison of some of the selected methods for our research. The performance of all the methods are thoroughly investigated in our research, which may not have been done in scRNA-seq data before. The compelling rationale for our research is to fill this gap in knowledge and form a general recommendation on the performance of these methods. Furthermore, we propose a new method (an algorithm for feature selection using penalized regression methods) that uses fewer genes to execute the best of the selected methods and improve its prediction performance measured via AUC. This study is an extension of the research presented at the Comparative Genomics: 19th International Conference, RECOMB-CG 2022, La Jolla, CA, USA, 20–21 May 2022, Proceedings [26]. The R codes for the analysis are provided in Appendix A.

The rest of this article is organized as follows. Section 2 explains different methods and metrics used in this study. Section 3 introduces the scRNA-seq data sets and the research design. Section 4 and Section 5 showcase the results, discuss the findings and their biological interpretations, and briefly introduce potential future directions for this research.

## 2. Methods

To reach our objectives, we used several penalized regression methods in this paper. These regression methods and the performance metrics suitable for this study are described below.

### 2.1. General Representation of Penalized Regression

Penalized regression methods select important feature variables available in high-dimensional data obtained from scRNA-seq technologies by producing sparse solutions that may result into better predictive models by simplifying the expression using a limited number of genes. Mathematically speaking, a penalized method solves an optimization criterion, which usually comprises two components: a loss function plus a penalty term. Let the regression equation be
(1)Y=Xβ+ϵ,
where *Y* is a *n*-vector of response variable, X is a n×p matrix for the predictor variables, β is a *p*-vector for the regression coefficients, and $\epsilon ∼N(0,\sigma^{2}I)$ is the random error term. We consider that both X and *Y* are centered and scaled in Equation (Equation 1). The penalized regression method estimates the regression coefficients by minimizing the penalization criterion as follows:(2)$$\hat{\beta }=\underset{\beta }{\mathrm{argmin}}(\frac{1}{n}||Y-{X\beta ||}^{2}+\lambda ||\beta ||),$$
where ||β|| is the norm of coefficient vector β and λ≥0 is the tuning parameter to be optimized using cross-validation. The first term $||Y-{X\beta ||}^{2}/n$ is called the loss function and the second term λ||β|| is the regularization term often called penalty. The main difference between different penalized regressions methods stems from different forms of regularization used as the penalty. The most frequently used L1 and L2 norms are the popular choices for the penalty [12], which are defined as
(3)$$L1\;norm={\left||\beta |\right|}_{1}=\sum_{i=1}^{p}\left|\beta_{i}\right|,$$
(4)$$L2\;norm={\left||\beta |\right|}_{2}^{2}=\sum_{i=1}^{p}\beta_{i}^{2},$$
respectively.

#### 2.1.1. Ridge Regression

Ridge Regression is a penalized regression method where the penalty term is the sum of squared coefficients (Equation (Equation 4)). Regression methods usually suffer from ill conditioning due to high correlation between predictor variables. The method of ridge regression appears handy, reducing the large absolute coefficients towards zero, a problem incurred from the high correlation between predictors. Ridge regression has been used to model gene expression data in many studies [27,28,29]. The estimates of the regression coefficients obtained from ridge regression are expressed as
(5)$${\hat{\beta }}_{Ridge}=\underset{\beta }{\mathrm{argmin}}(\frac{1}{n}||Y-{X\beta ||}^{2}+{\lambda ||\beta ||}_{2}^{2}),$$
where the penalty term ${\left||\beta |\right|}_{2}^{2}$ is the L2 norm of the regression coefficients.

#### 2.1.2. Least Absolute Shrinkage and Selection Operator Regression

The Least Absolute Shrinkage and Selection Operator (lasso) regression [13] minimizes the residual sum of squares subject to the constraint of the sum of the absolute coefficients being less than the tuning parameter. As obvious from the above sentence, lasso uses L1 norm (Equation (Equation 3)) as the penalty. Compared with the ridge regression, which can only shrink coefficients towards 0, lasso can make some of the coefficients exactly equal to zero, thereby producing a simpler model by including only those predictors for which the coefficients are not zero. By ignoring feature variables for which the coefficients are zero, lasso produces a more interpretable model. The estimator of lasso regression is is obtained as
(6)$${\hat{\beta }}_{Lasso}=\underset{\beta }{\mathrm{argmin}}(\frac{1}{n}||Y-{X\beta ||}^{2}+{\lambda ||\beta ||}_{1}),$$
where the penalty term ${\left||\beta |\right|}_{1}$ is known as the L1 norm of the regression coefficients.

#### 2.1.3. Elastic Net Regression

Elastic net regression, proposed by [14], is a method of estimating regression parameters by regularizing the coefficients using a convex combination of the L1 and L2 norms. As a result of using both of the L1 and L2 norms, the elastic net regression enjoys the properties of both ridge and lasso. Using elastic net, a large regression coefficient can be shrunk towards zero while forcing some of the other regression coefficients to becoming exactly zero. The estimates of the regression coefficients using elastic net can be expressed as
(7)$${\hat{\beta }}_{EN}=\underset{\beta }{\mathrm{argmin}}(\frac{1}{n}||Y-{X\beta ||}^{2}+\lambda_{1}{\left||\beta |\right|}_{1}+\lambda_{2}{\left||\beta |\right|}_{2}^{2}).$$

The L1 norm part of the penalty generates a sparse model and the L2 norm part of the penalty encourages greater shrinkage to large coefficients [30].

#### 2.1.4. Group LASSO

Yuan and Lin [17] proposed group lasso, which allows selecting subsets of important variables. Unlike the lasso, which selects individual variables, group lasso select groups of feature variables in the form of subsets. The selection of subsets of features is useful in scRNA-Seq data as one may wish to include or exclude the subsets of genes that are associated with a pathway related to an outcome, which is practically important to an application. Let j∈{1,2,⋯,J} represent the indices for the groups of variables and *n* be the number of observations. For each group *j*, let $X_{j}$ be $n\times p_{j}$ submatrix of X with columns corresponding to predictor variables in group *j* and $\beta_{j}$ be the corresponding coefficient vector of length $p_{j}$. Then, the regression equation for the group lasso can be written as
(8)$$Y=\sum_{j=1}^{J}X_{j}\beta_{j}+\epsilon .$$

Here, the vector of regression coefficients is $\beta ={\left(\beta_{1}^{\prime },\beta_{2}^{\prime },\cdots ,\beta_{J}^{\prime }\right)}^{\prime }$, and the design matrix can be written as $X=(X_{1},X_{2},\cdots .X_{J})$. When $X_{j}^{\prime }X_{j}=I_{p_{j}}\;\forall j\in \left\{1,2,\cdots ,J\right\}$, the above regression equation simplifies to Equation (Equation 1). For a symmetric and positive definite kernel matrix $K_{j}=p_{j}I_{p_{j}}$, the estimates of the regression coefficients from group lasso can be expressed as
(9)$${\hat{\beta }}_{GL}=\underset{\beta }{\mathrm{argmin}}(\frac{1}{n}||Y-\sum_{j=1}^{J}X_{j}\beta_{j}{\left|\right|}^{2}+\lambda \sum_{j=1}^{J}\left|\right|\beta_{j}^{{}^{\prime }}K_{j}\beta_{j}{\left|\right|}^{\frac{1}{2}}),$$
where λ≥0 is the penalty coefficient.

#### 2.1.5. Sparse Group LASSO

One possible limitation of group lasso is that while it selects a collection of subsets of features, all of the estimated regression coefficients in a selected subset of features will be nonzero. That means there is no variable selection within a selected subset or group of features. Often, both sparsity of subsets and features within each subset might be important in an application. In scRNA-seq application, identifying important genes in the biological pathways may be of interest. Simon et al. [18] proposed sparse group lasso as a potential improvement to this problem. They proposed a method to estimate the regression coefficients using a sparse group lasso as follows:(10)$${\hat{\beta }}_{SGL}=\underset{\beta }{\mathrm{argmin}}(\frac{1}{n}||Y-\sum_{j=1}^{J}X_{j}\beta_{j}{\left|\right|}^{2}+\lambda \sum_{j=1}^{J}\left|\right|\beta_{j}^{{}^{\prime }}K_{j}\beta_{j}{\left|\right|}^{\frac{1}{2}}+{\alpha \lambda ||\beta ||}_{1})),$$
where the regularization parameters are λ≥0 and α∈[0,1]. Therefore, the penalty term in the sparse group lasso is a convex combination of the penalty terms in lasso and group lasso. Note that when α=0, the SGL reduces to the group lasso. Further, when α=1, the model reduces to lasso. This feature of sparse group lasso is analogous to elastic net, which is a convex combination of ridge and lasso regression.

#### 2.1.6. Drop LASSO

The modeling of scRNA-seq data may become complicated and challenging to create a number of modeling and computational problems. For example, as there is a little amount of RNA is present in each cell, a big proportion of polyadenylated RNA might be stochastically lost during sample collection including cell lysis, reverse transcription, or amplification. As a result, a good collection of genes may fail to be detected even when they are expressed, introducing a type of error referred to as dropouts. This may introduce a lot of zeros in the scRNA-seq data. The presence of excess zeros in the data might pose considerable challenges to the analysis and biological interpretation, and may give rise to new stochastic models for data shifting and scaling, and visualization or gene differential analysis. As a remedy, Khalfaoui and Vert [12] proposed drop lasso to arrive at a solution to the above problem. Drop lasso is a combination of the dropout regularization proposed by [13,31]. The methodology creates a sparse linear model that appears to be robust to the noise by artificially augmenting the training set with new examples. First, a random permutation of rows is performed in matrix X with *n* observations and *p* predictor variables. After that, each of the permuted rows in X undergo an element-wise scaling with a random dropout mask—a vector of 1’s and 0’s —of length *p* to create a new matrix $X_{drop}$. The drop lasso estimator then optimizes the following:(11)$${\hat{\beta }}_{DL}=\underset{\beta }{\mathrm{argmin}}(\frac{1}{n}||Y-X_{drop}{\beta ||}^{2}+{\lambda ||\beta ||}_{1}),$$
where the regularization parameter λ≥0 is also known as the tuning parameter.

#### 2.1.7. Big LASSO

Zeng and Breheny [19] implemented big lasso (in R), which can handle large-scale, ultra-high-dimensional data. Their approach incorporates out-of-core computation continuously by importing data into computer memory only when it is required. This is conducted with the support of memory-mapped files, which have the ability to store huge amounts of data on the computer disk. big lasso also incorporates efficient feature selection, which can make the computation much faster. The prime differences between big lasso and other regular lasso methods are in their ability to do out-of-core parallel computation. The estimator of the coefficients of big lasso can be obtained as
(12)$${\hat{\beta }}_{BL}=\underset{\beta }{\mathrm{argmin}}(\frac{1}{n}||Y-{X\beta ||}^{2}+{\lambda ||\beta ||}_{1}),$$
where the tuning parameter is λ≥0, which controls the amount of regularization to be injected.

### 2.2. Clustering

Clustering is the process of grouping data instances into clusters so that the instances in the same cluster have high similarity [32]. Two popular clustering methods in machine learning used in this study are hierarchical and *K*-Means clustering. They are briefly described below.

#### 2.2.1. Hierarchical Clustering

In hierarchical clustering, the data values are placed into groups or clusters hierarchically [32]. The resulting clusters of data values are displayed in a tree-like diagram called a dendrogram. The resulting dendrogram, when displayed graphically, appears to be useful in determining the optimal number of groups or clusters. In this study, we use hierarchical clustering to group feature variables, specifically to group genes into clusters, before sending them down to group lasso or SGL. The ability to group genes using hierarchical clustering bypasses the requirement of domain-specific knowledge of genes to use prior groups.

#### 2.2.2. *K*-Means Clustering

As in hierarchical clustering, the *K*-Means clustering algorithm also clusters data values into groups. The *K*-Means algorithm starts by randomly selecting *k* centroids of *p*-dimensional feature space [33]. The initial values of *k* are often supplied by the user. After that, the clusters are formed by repeatedly assigning data values into their nearest centroid. The algorithm continues iteratively by recalculating the centroids in such a way that the within cluster sum of squared error is minimized. Often, regularization is performed to determine the optimal number for *k*, which may improve prediction accuracy [34]. In this study, we applied K-means clustering to cluster cells at the final step of the algorithm.

### 2.3. K-Fold Cross-Validation

Cross-validation (CV) is a method of randomly dividing data values into folds, which we often call training and test folds, to calculate prediction performances of the models and to tune model parameters [35]. In *k*-fold CV, the data values are first randomly divided into *k* folds, of which k−1 of them are used to train a model and the rest are used to test the model. The test performances are then calculated for each of the *k* random folds separately by swapping the folds for a total of *k* times. The average prediction performance from the *k* test folds is used as the overall prediction performance of the model. In this study, we used 10-fold CV by assigning k=10. As mentioned before, this 10-fold CV is also used to tune the model hyperparameters.

### 2.4. Evaluation Metrics

To evaluate the performance, the methods are compared using the performance metrics calculated from a confusion matrix [36] shown in Figure 1. True Positives (TP) are the positive cases that are predicted positive by the ML model. Similarly, True Negatives (TN) are the negative cases that are predicted negative. False Positives (FP) are the negative cases that are incorrectly predicted as positive by the ML model. Likewise, False Negatives (FN) are the positive cases that are misclassified as negative by the ML model.

#### 2.4.1. Sensitivity

Sensitivity, also known as recall or True Positive Rate (TPR), is the proportion of positive cases that are predicted positive out of all positive cases. This metric is useful to detect positive cases when the classes are imbalanced [37].
$$Sensitivity=\frac{TP}{TP+FN}.$$

#### 2.4.2. Specificity

Specificity is the proportion of negative cases that were predicted as negative out of all negative cases. False Positive Rate (FPR), calculated as
FPR=1−Specificity,
is complementary to specificity. FNR is not sensitive to the changes in data distributions and can be used with imbalanced data [37].
$$Specificity=\frac{TN}{FP+TN}.$$

#### 2.4.3. Area under the Curve (AUC) of Receiver Operating Characteristic Curve (ROC)

An ROC is a two-dimensional plot that assigns the true positive rate (TPR) on the *y*-axis and the false positive rate (FPR) on the *x*-axis [38]. Often, ROC is used to evaluate the performance of a classifier in a binary classification problem. The AUC values under an ROC curve evaluate the ranking performance of a classifier. In an unbalanced binary classification problem, the AUC is a better assessment metric than accuracy. However, the computational complexity of AUC is a bit higher than that of the other evaluation metrics [39].

### 2.5. Proposed Algorithm for Feature Selection Using Penalized Regression Methods

In this paper, we propose a novel algorithm for feature selection using 5 penalized regression methods (ridge, lasso, elastic net, drop lasso, and SGL). The proposed algorithm aims to handle the ’large-p-small-n’ problem of scRNA-seq data through step-by-step feature selection, as shown in Figure 2. The algorithm is designed as a sequential execution of data preprocessing, feature selection, clustering, and visualization. In the data preprocessing stage, cell groups are assigned to class 0 or class 1 as per the labels in the raw data. Then, to randomize the data points, the cells are shuffled within each class. In the next step, all the genes with no variability in expression for all the cells are removed. The data are then split as test (10% of the data) and training (90% of the data) sets for a 10-fold cross-validation. The training data are then used as the input for the next stage. These conclude the data preprocessing stage. Next, our algorithm uses several penalized regression methods for feature selection in two stages. The first stage of feature selection is carried out by ridge, lasso, elastic net, and drop lasso methods. Each algorithm selects genes that appear to be important. Here, the genes with coefficients above the mean of the absolute value of coefficients are selected. The union of all the top important genes from the 4 penalized regression methods is used as the input for the next stage. The union of top important genes contains less than the 50% of the genes in the original data. This reduced subset of genes is grouped using Hierarchical Clustering. Then, SGL is applied using the groups of important genes from Hierarchical Clustering. The resulting model with the selected set of genes is applied on the test data for prediction. As we have used 10-fold CV, this process is repeated for a total of 10 times. The algorithm repeats the process of selecting important genes using the SGL model coefficients. The selected genes are then visualized in a gene vs. coefficient plot. This final set of genes is then used to cluster cell types in the scRNA-seq data using the *K*-Means clustering. The steps to implement the proposed algorithm are presented in Algorithm 1.
**Algorithm 1:** Steps to implement the proposed algorithm1.Load data set into R and assign classes 1 and 0 to the two selected group of cells to form a binary classification problem.2.Shuffle cells within each class to randomize the data points.3.Remove genes with no variability in expression across all cells.4.Split the data set into training (90%) and test (10%) for 10-fold cross validation.(a)Fit ridge, lasso, elastic net, and drop lasso.(b)Find the top important genes from each method. The top genes are the genes that have coefficients above a cut off (mean of absolute value of coefficients).(c)Form a gene pool by taking union of the top important genes from the 4 models; for instance, Figure 3 and Figure 4 represent the gene pool of data sets GSE123818 and GSE71585, respectively.(d)Fit SGL with the new gene pool pre-grouped by hierarchical clustering.(e)Save the coefficients of SGL.(f)Repeat the steps for a 10-fold CV.5.Calculate the average of coefficients for each gene across the 10 folds and sort the genes.6.Visualize the gene versus coefficients plot and select the final set of genes using an elbow curve.7.Cluster all the cells by applying K-means clustering on the top important genes.

**Figure 2 biology-11-01495-f002:**
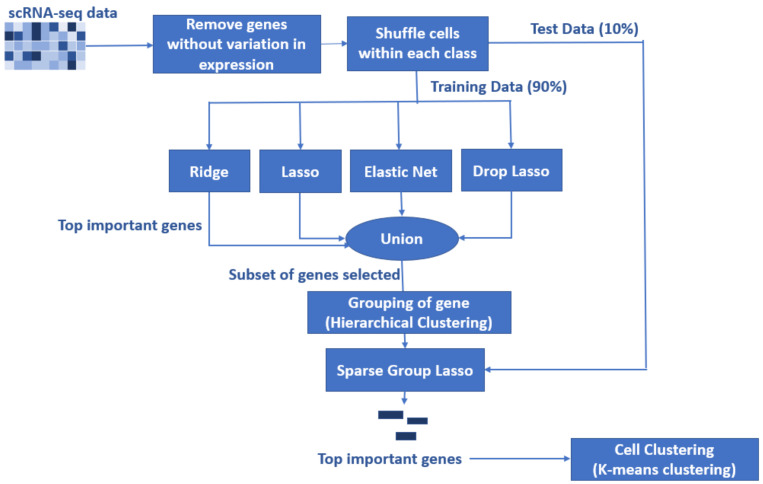
Schematic of the proposed algorithm for feature selection using penalized regression methods. There is a considerable reduction in the number of genes prior to the execution SGL. The top important genes selected by SGL are used to cluster cell groups using K-means clustering.

**Figure 3 biology-11-01495-f003:**
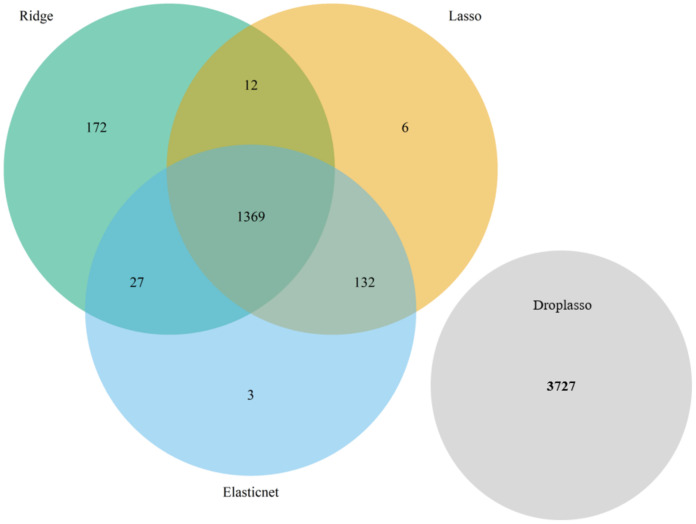
The gene pool of data set GSE71585 formed by taking the union of top important genes from the ridge, lasso, elastic net, and drop lasso. For this data set, the top important genes from drop lasso had no intersection with the top genes from other 3 methods. In the proposed algorithm, gene pool is formed with the union of the top important genes from the 4 methods rather than an intersection because there may not always be an intersection due to differences in regularization used.

**Figure 4 biology-11-01495-f004:**
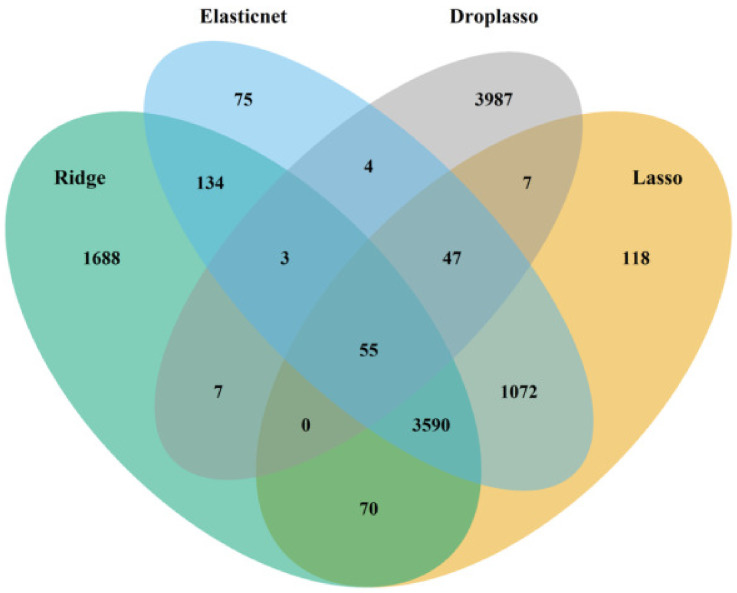
The gene pool of data set GSE123818 formed by taking the union of top important genes from ridge, lasso, elastic net, and drop lasso.

## 3. Experimental Data and Research Design

### 3.1. Experimental Data

There are a total of 5 data sets used in this study. To compare the penalized regression methods, 4 scRNA-seq data sets from 3 different species (Human, mouse, and plant) were used. One additional data set was used to compare the performance of the proposed method with the best-performing single method (SGL). This study used real data sets rather than simulated data sets because real data sets often contain variables with complex interrelationships that are difficult to reproduce using a simulation. Two data sets were downloaded from Conquer (http://imlspenticton.uzh.ch:3838/conquer/ accessed on 1 July 2021), a curated database of continuously processed, analysis-ready, and well-documented scRNA-seq data. There are over 40 publicly available scRNA-seq data sets on this website, and they all have the count and transcripts per million (TPM) estimates for genes, quality control, and exploratory analysis reports. The remaining three datasets (GSE123818, GSE81861, and GSE157997) are all available from the Gene Expression Omnibus (GEO) and can be retrieved using the accession numbers. We summarize these data sets in Table 1 including the number of genes and cells, GEO accession numbers, organism, technology used, and the source links. In our study, the cells were separated into two groups (classes) as per the label from each of the original experiments. The data set GSE81861 had batch effect correction in the original study. For the other data sets, it is not clear if batch effect correction was carried out from the descriptions of the original experiments. No additional batch effect correction was conducted in the data preprocessing stage for this research. We did not consider any other biological effects, which may be considered a potential limitation of this study.

#### 3.1.1. Gene Expression Data from Mice

The Mus musculus data set GSE60749 was generated by single-cell expression profiling of PSCs under different chemical and genetic perturbations [40]. In the original study, gene expression variability in Pluripotent Stem Cells (PSCs) was investigated and their expression levels were quantified as transcripts per million reads (TPM). In our study, 183 individual v6.5 Mouse Embryonic Stem Cells (mESCs) and 84 Dgcr8 −/− mESCs that lack mature miRNAs (knockout of a miRNA processing factor) were selected as two classes. The 183 cells were labeled as class 1 and the 84 cells as class 0. There were 22,443 genes in the original data. After removing the genes with no expression variance across all cells, we had 15,508 genes in total.

The Mus musculus data set GSE71585 was generated by scRNA-seq of adult mouse primary visual cortex in a study [41] conducted to understand cell type diversity in the nervous system. The original data set contains 1809 cells and 24057 genes, of which 79 Ntsr1_tdTpositive_cell were labeled as class 0 and 57 Ntsr1_tdTnegative_cell as class 1. After conducting a similar process as in the previous data set (removing genes with no expression variance), we had 17,870 genes in total.

#### 3.1.2. Gene Expression Data from Plant

The Arabidopsis Thaliana data set GSE123818 came from the study of Spatiotemporal Developmental Trajectories in the Arabidopsis Root [42]. The original study was to investigate Arabidopsis root cells in different developmental fates and times. Using Illumina NextSeq technique, the authors generated mRNA profiles of 6-day-old wild-type (wt) and shortroot-knockout (shr) Arabidopsis thaliana roots by deep sequencing of single-cell and bulk RNA libraries (wild type only), in duplicate (bulk and wild-type single cell) and singlicate (shr-3). The original data set contains 4727 wt cells and 1099 shr cells. Since there are significant differences in the numbers of cells of the two types, we selected all the 1099 shr cells and randomly selected 1099 cells from the total 4727 wt cells to create a balanced data set for our study. Then, after removing the genes with no variance in expressions, we had 24,075 genes in total.

#### 3.1.3. Gene Expression Data from Human

The data set relating homo sapiens GSE81861 is from the analysis of transcriptional heterogeneity in colorectal tumors [43]. Intratumoral heterogeneity is a major obstacle to cancer treatment and a significant confounding factor in bulk-tumor profiling. A study [43] was conducted on the transcriptional heterogeneity in colorectal tumors and their microenvironments using scRNA-seq. The original data set contains 272 primary colorectal tumor cells and 160 matched normal mucosa cells. The 272 primary colorectal tumor cells are labeled as class 1 and 160 matched normal mucosa cells as class 0. With the shape differences of the original data compared with other data sets, this data set was then transposed to form a matrix of 432 rows (cells) and 57,241 columns (genes). After removing all the genes with no variance in expressions among all cells, we had 38,090 genes in total.

The last data set GSE157997 belongs to studies relating homo sapiens and Mus musculus. This data set is a super series of the subseries GSE157995 and GSE157996. From the subseries, GSE157996 (homo sapiens) was selected for further analysis. This data set was generated from an experiment to investigate human lung epithelial cell subpopulation between healthy donors and idiopathic pulmonary fibrosis (IPF) patients [44]. IPF is a fatal lung disease with life expectancy of 3–5 years. There were 6 healthy donors and 6 IPF Patients. The cells from each subject were labeled separately. Samples of normal donor 1 and IPF patient 1 were selected for analysis in this study. For this data set, the two groups of cells were collected and sequenced separately from two different people. The plan is to find the topmost differentiated genes between the healthy donor and IPF patient. As binary classification is studied in this research, any subpopulations in the Lin–EpCAM+ cells are not considered for further analysis. In this data set, there are 33,694 genes in the gene expression profile. There are 2788 IPF patient cells and 1504 healthy donor cells. A total of 1504 cells were randomly sampled from IPF patient cells to make the two classes more balanced. Healthy donor samples were assigned the value of 0 and IPF patient samples were assigned the value of 1. All the genes that have the same value across all the cells were removed as a part of data preprocessing, which reduced the number of genes to 24,057.

### 3.2. Research Design

We started the analysis by preprocessing the data sets to make sure they were compatible for use in different R packages used. Data sets are available as a single CSV file or multiple files with genes as rows and cells in the columns. Multiple files were combined into a single file and transposed to obtain a single matrix with cells as rows and genes in the columns.

In this research, we focused on binary classification. Therefore, two cell groups were created and labeled as classes 0 and 1 for each experimental data set. Cells were also shuffled within each class and for the data sets with significant differences in the numbers of samples for the two classes. Therefore, we balanced the data sets before conducting analysis. For all the data sets, genes with no expression variance across all cells were removed. We then verified how each method performs in terms of AUC, computational time, and cross-validation with the different scRNA-seq data used. Here, a 10-fold stratified cross-validation was performed to calculate the performance metrics. We used different hyperparameters for each method, which are shown in Table 2. We used hierarchical clustering for grouping variables prior to sending down to group lasso and sparse group lasso.

After calculating the performance metrics, the best-performing methods were selected and combined to form the proposed algorithm for feature selection using penalized regression methods. Finally, the performance metrics of the proposed algorithm were compared with those of the top-performing method. Figure 2 illustrates the proposed algorithm for feature selection using penalized regression methods. First, we used 4 data sets (GSE60749, GSE71585, GSE81861, and GSE123818) for benchmarking the penalized regression methods. Second, the fifth data set GSE157997 was used to test results of the proposed method on data not used during benchmarking methods. Note that we could extend the study to include more cell groups and verify methods for the multi-classification problem in the future; however, in the current study, we focus on comparing the methods for binary classification only.

#### Hardware Requirements

All the experiments are conducted on an Ubuntu 20.04.4 LTS (GNU/Linux 5.4.0-100-generic x86_64) virtual machine, hosted by Compute Canada, with 32 GB RAM. The programming language used is R Project for Statistical Computing version 4.1.2 [46].

## 4. Results

### 4.1. Benchmarking Penalized Regression

The first objective of this study is to compare the performance of the selected penalized regression methods. The results in terms of cross-validated AUC (CV-AUC) and computational time are shown in Table 2 and Table 3, respectively. Figure 5 shows the CV-AUC by method for the four data sets. From Table 2 and Figure 5, we observe that the top 5 methods in order of importance are SGL, grplasso, droplasso, biglasso, and lasso. As evident from Table 4, the variance of CV-AUC is close to 0 for all methods when rounded to 2 decimal points. Notice that SGL and grplasso outperform the other methods in terms of CV-AUC, whereas ridge regression method has the least CV-AUC. This could be because grplasso and SGL incorporated grouping of genes into the model, whereas ridge regression treats all the genes equally. After performing Friedman test (a non-parametric statistical test) with the CV-AUC results of the 7 methods, we found statistically significant differences between their performance at *p*-value = 15%. An Analysis of Variance (ANOVA) test such as the Friedman test alone cannot reveal which of these methods has a significant difference in performance [47]. Therefore, we use a post-hoc test in the next step. Post-hoc tests are statistical tests carried out on the subgroups in data already tested in other analyses such as an ANOVA test [47]. A Nemenyi test is one such post-hoc test. A Nemenyi test on our results revealed that the difference in performance is between SGL and ridge regression. However, more data sets may need to be analyzed to further verify these results.

After comparing the computation time of the methods given in Table 3, it is evident that the SGL method has the least average computation time across the data sets. Ridge regression requires the highest computation time on average across the four data sets. This is because of its computational complexity arising from using all of the features (genes) in the data. SGL can deselect an entire group of genes or some of the genes within a group, which reduces the computational complexity. Data set GSE60749 has lesser computation time than GSE81861 across all methods because it has fewer non-zero coefficients. Average computation time for lasso, elastic net, and big lasso increased with an increase in the number of samples in the data.

### 4.2. Performance of the Proposed Algorithm for Feature Selection

The second objective of this study was to combine the top-performing penalized regression methods to improve predictive AUC and perform gene selection. From the discussion of the results of the first objective, we observed that SGL and grplasso were better candidates to form a new method. In terms of gene selection, SGL outperforms grplasso. SGL could identify the top important genes in one fold of 10-fold CV, whereas grplasso takes multiple folds to obtain the same result. In other words, the results of top important genes changed from fold to fold of the 10-fold CV for group lasso. Therefore, SGL was chosen over grplasso for the proposed algorithm. SGL achieves better AUC than biglasso in comparable time for data sets of size 63 MB to 252 MB when tested on a computer with a 32 GB processor. Since biglasso did not show a significant improvement in computation time relative to SGL, biglasso is not included in the proposed method.

To develop the proposed algorithm using penalized regressions, we selected the ridge, lasso, elastic net, and droplasso to form a filter that produces a gene pool with the number of genes reduced considerably relative to all the available genes. The gene pool was formed by taking a union of the top important genes selected from the four methods as the top important genes have some variations between methods. A union of top important genes is, therefore, more likely to capture the important and differentially expressed genes. The gene pool thus formed is grouped and sent down to SGL to calculate predictive AUC.

The experimental data used to test the performance of the proposed method include all four data sets in the benchmarking, and also a new independent data set (GSE157997). This data set has the largest number of cells among all five data sets, and was not used to develop the proposed algorithm. It is therefore expected to generate a fair and independent comparison between the proposed method and others.

In the proposed algorithm for feature selection using penalized regression methods, feature selection is performed in two stages. The first stage creates a gene pool that has a significantly reduced number of genes (less than 50%) compared to the original data as shown in Table 5. Then, the gene pool is grouped using hierarchical clustering and the grouping information is used as input to SGL.

From our experiments, the hierarchical clustering and SGL originally required a 32 GB RAM processor for the computation. With the contribution of the proposed method, large numbers of genes are reduced in the gene pool. We notice that the proposed method can be executed on an 8 GB RAM processor for the following data sets: GSEGSE60749 and GSE71585. The proposed algorithm can be used for other high-dimensional data sets and feature selection as well. As shown in Table 5, this method consistently improves the predictive AUC of SGL. The genes in the SGL model are visualized in a gene versus coefficients plot to select the final subset of genes. Then, this subset of genes is used for cell clustering of each data set using *K*-Means clustering.

Essentially, the proposed algorithm tackles the “large-p-small-n” problem by applying feature selection, on the high-dimensional scRNA-seq data, in a sequential fashion. At each stage, different penalized regression methods are used for feature selection. The first stage (ridge, lasso, elastic net, and drop lasso) reduces the number of genes to less than 50% of the original size, and the second stage (SGL) reduces it further to make the number of genes comparable to the number of cells.

## 5. Discussion

In this section, we discuss the final subset of genes selected by the proposed method and the cell clustering within each data set to reveal potential biological interpretation and meanings.

### 5.1. Mus Musculus Dataset GSE60749

Figure 6 shows the genes versus coefficients plot for the first data set GSE60749 (mESCs). The most important genes identified by the proposed method are 44441, 44260, 44454, 44446, 44450, 44440, Pbld, Lifr, Hist2h4, and AK203176 (Table 6). As in Figure 7, the data set GSE60749 after clustering by the top important genes has well-separated classes. In addition, we notice that the top genes (44441, 44260, 44454, and 44440) turned out to be Piwi-interacting RNAs (piRNAs). piRNAs are non-coding RNAs whose function is largely unclear to the biomedical field. According to a study by [48], these RNAs are abundant in testes and have interaction with a murine Piwi protein. This study also suggested that piRNAs could be potentially involved in gametogenesis due to their abundance in germline cells and murine Piwi protein mutation causing sterility in males. A study [49] published in 2011 investigated the relationship between piRNAs and carcinogenesis, and identified piRNAs as a potential marker for cancer diagnosis. Even though the exact function of these piRNAs is unknown, the available study proves that the proposed algorithm selected the most relevant genes. The next two genes Pbld and Lifr are also involved in pertinent biological pathways. The Pbld gene is linked to negative regulation of transforming growth factor beta receptor signaling pathway and Lifr is linked to the ESC pluripotency pathway. The original study by [40] for the data set GSE60749 also suggests that Lifr is a marker gene.

### 5.2. Mus Musculu Dataset GSE71585

The genes versus coefficient plot for the second data set GSE71585 is shown in Figure 8 and their functions are listed in Table 7.

The proposed algorithm for feature selection can cluster the primary visual cortex cell groups well (Figure 9) with the topmost important genes. The final genes selected are Calm2, Snap25, 0610005C13Rik, and 0610007C21Rik. This data set contains Ntsr1 (neurotensin receptor 1) tdT (tdTomato—an exceptionally bright red fluorescent protein) positive cells and Ntsr1 tdT negative cells. Calm2 gene is active in the cortex, frontal lobe, and a few other organs. It enables N-terminal myristoylation domain binding, calcium ion binding, and protein binding. This gene is also involved in some important pathways such as Alzheimer’s disease and Glycogen Metabolism. From the information published on NCBI, we found that several infants with severe forms of the long-QT syndrome (LQTS) who displayed life-threatening ventricular arrhythmias together with delayed neurodevelopment and epilepsy were found to have mutations in either this gene or another member of the calmodulin gene family [50]. The second gene, Snap25, is also an important gene that is involved in 10 biological pathways. Snap25 is a known pan-neuronal marker gene as per the original study [41]. A variety of disorders such as attention deficit hyperactivity disorder, schizophrenia, and type 2 diabetes mellitus are studied using this gene as shown in NCBI [51]. The relationship between knockout of tdT protein in Ntsr1 cells and the genes selected by the proposed method warrants further examination.

### 5.3. Arabidopsis Thaliana Dataset GSE123818

The most important genes for the data set GSE123818 found by the proposed method are AT2G43610, AT4G05320, AT2G07698, and AT3G51750 (Figure 10). This data set has more overlap between cell clusters compared to that of the other data sets (Figure 11). Table 8 lists the final set of selected genes and their functions. The first gene, AT2G43610, is a gene coding Chitinase family protein, and it acts upstream of or within the root development process of Arabidopsis thaliana [52]. It is involved in the Chitin degradation II pathway. The AT2G43610 gene was reported as a marker gene by a study [53] on cell-cycle-regulated gene expression in Arabidopsis. The second gene AT4G05320, also known as Polyubiquitin 10, is one of five polyubiquitin genes in Arabidopsis thaliana. These genes encode ubiquitin protein that covalently attaches to substrate proteins for their targeted degradation [54]. The AT2G07698 gene is expressed during the seed and seedling development stages and involved in proton transmembrane transport [55]. The gene AT3G51750 encodes a protein that acts upstream of or within the root development process [56]. From the results, we see that the proposed method has selected genes that are involved in the development of the root cells. Our findings further support the study of these genes regarding the development of root cells in Arabidopsis Thaliana.

### 5.4. Homo Sapiens Dataset GSE81861

The set of selected genes by the proposed method in the GSE81861 data set are shown in the genes versus coefficients plot (Figure 12). For GSE81861, cell groups are clustered with some overlap in classes (Figure 13). Table 9 lists the final set of selected genes and their functions. The most important genes are FABP1, SAT1, PHGR1, LGALS4, FRYL, MT1E, HSP90AA1, and HNRNPH1. The first two genes, FABP1 and SAT1, are important and each is involved in 13 different biological pathways. FABP1 is involved in fatty acid uptake, transport, and metabolism. It is also found in Peroxisome proliferator-activated receptor (PPAR) signaling pathway, PPAR-alpha pathway, fatty acid transporters, and metabolism [57]. The FABP1 gene has also been identified as a novel prognostic marker for gastric cancer by a study [58] recently published in June 2022. The second gene SAT1 enables N-acetyltransferase activity and spermidine binding. It participates in pathways such as metabolism, spermine and spermidine degradation I, and NOTCH1 regulation of endothelial cell calcification pathway. Defects in this gene are associated with keratosis follicularis spinulosa decalvans (KFSD) [59]. The third gene LGALS4 codes the galectin proteins. Galectins are a family of beta-galactoside-binding proteins implicated in modulating cell–cell and cell–matrix interactions. The LGALS4 gene is underexpressed in colorectal cancer [60]. The fourth gene HSP90AA1 is found in 115 biological pathways such as signaling by EGFR, EGFRvIII, and ERBB2 in Cancer [61]. The HNRNPH1 gene is identified in 12 biological pathways. This gene may be involved in hereditary lymphedema type I. Knockdown of heterogeneous nuclear ribonucleoprotein H1 (HNRNPH1) by siRNA inhibits the early stages of HIV-1 replication in 293T cells infected with VSV-G pseudotyped HIV-1 [62].

Looking at Table 9, we observe a remarkable connection between the topmost important genes and HIV-1. We note that four genes (SAT1, MT1E, HSP90AA1, and HNRNPH1) from a colorectal tumor tissue have interactions with HIV-1 proteins. Cancer and HIV-1 association has been subject to a variety of research [63,64,65,66]. These factors could potentially lead to the finding that the genes selected by the proposed algorithm are pertinent to human colorectal cancer.

### 5.5. Homo Sapiens Dataset GSE157997

The last data set GSE157997 is used to independently evaluate the performance of the proposed algorithm with others. In terms of AUC, the proposed algorithm performs the same as the best-performing single method (SGL) (Figure 5). The proposed method shows promising results in terms of gene selection as well. The genes versus model coefficients plot is shown in Figure 14. Table 10 lists the final set of selected genes and their functions. As shown in Figure 15, there is minimal overlap of cells in the two groups. The topmost important genes found with the new method are TMSB4X, S100A6, B2M, SEPW1, and HLAB-1. Notably, the TMSB4X gene is involved in cell proliferation, migration, and differentiation and is also linked to tumorigenesis. Several existing studies [67,68] have identified TMSB4X gene as a marker gene for cancer. The second gene S100A6 is involved in the up-regulation of fibroblast proliferation. The B2M gene is involved in 25 pathways including the immune system, as is the HLA-B gene, which also plays a central role in the immune system. Inflammation and immunity are also included in the pathogenesis of IPF [44]. Therefore, these genes are also related to IPF. Evidently, the proposed algorithm can select a highly relevant subset of genes.

The penalized regression methods select genes that are highly different in their expression between cell types. In our study, the penalized regression methods were evaluated for their ability to predict cell types based on the genes differentially expressed between them. Then, we proposed an algorithm using these penalized regression methods to select genes through successive stages of feature selection to improve cell type prediction performance via AUC. We have shown that, for all data sets, some of the top genes from the subset of differentially expressed genes selected by the proposed algorithm are known marker genes. We suggest using the genes identified by our proposed method as a pool of candidates for the future wet-lab experiments for verification.

## 6. Conclusions

This study benchmarks several state-of-the-art penalized regression methods on their performance to scRNA-seq classification. Based on the findings of the benchmarking, we proposed an algorithm of penalized regressions, which improved the prediction performance.

The results and findings show that the SGL method outperforms other methods in terms of predictive AUC and computation time. We note that the penalized regression methods can have many hyperparameters, and changes in these hyperparameters may affect the results. For instance, the number of groups of genes and the methods used for grouping the genes can cause differences in predictive AUC, computation time, and final gene selection by group lasso and SGL. The proposed algorithm for feature selection shows a better prediction compared with SGL.

The advantages of our proposed method are two-fold in the sense that it uses hierarchical clustering to find the grouping information of genes that bypasses the need to have much knowledge of genes in scRNA-seq data prior to the execution of SGL, and yet the differentially expressed genes selected by the proposed method and the cell clusters in the data have a strong association. It is because the proposed algorithm for feature selection carries out the feature selection by creating a gene pool based on the union of the top genes from four different methods. This step ensures that any relevant genes that may have been missed by any one of the penalized regression methods are highly likely to be captured by other methods and subsequently included in the gene pool. Due to the sequential nature of analysis and two stages of feature selection, the proposed algorithm may require more time to execute than SGL alone. However, the proposed method uses a smaller subset of genes as input for SGL, thereby reducing computational memory requirements from 32 GB RAM to 8 GB RAM.

This research has the potential to be extended to include more methods and relevant R packages, such as the Seagull [69], which also implements lasso, group lasso, and sparse group lasso. The proposed method sometimes can be time consuming as it is implemented in a sequential manner. Therefore, further reduction in computational time is possible via parallel computing. The use of other possible machine learning methods such as XGBoost [70] on top of SGL is a possible research direction worth exploring. In this study, we have explored two groups of cells. Any subpopulations inside the two groups are not investigated. Yet, scRNA-seq data sets may have more than two groups of cells. Therefore, the use of other R packages such as msgl [71], which can implement the multinomial classification method, is a potential future work to be explored.

## Figures and Tables

**Figure 1 biology-11-01495-f001:**
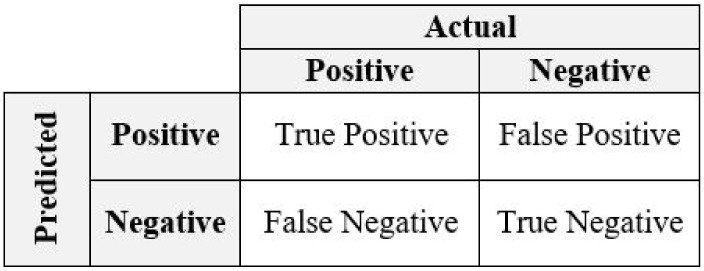
A Confusion Matrix from which performance metrics are usually calculated.

**Figure 5 biology-11-01495-f005:**
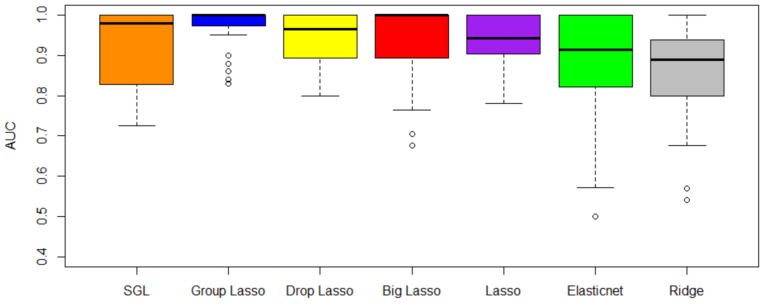
Average Cross-validation AUC across all four data sets. Even though group lasso has better AUC than SGL, SGL is better in terms of gene selection. Selection of the differentially expressed genes is of more importance for a scRNA-seq application than better prediction.

**Figure 6 biology-11-01495-f006:**
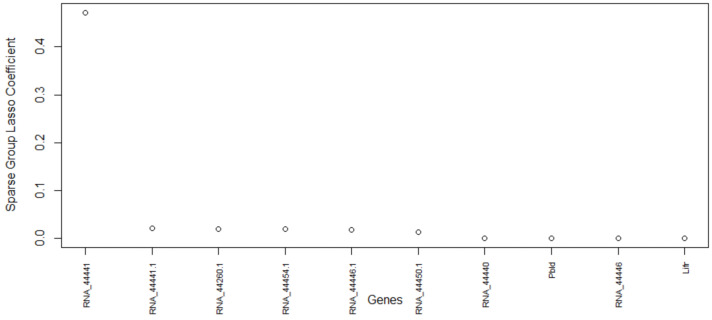
Genes versus coefficients plot for the data set GSE60749. Here, piRNA 44441 is the top important gene.

**Figure 7 biology-11-01495-f007:**
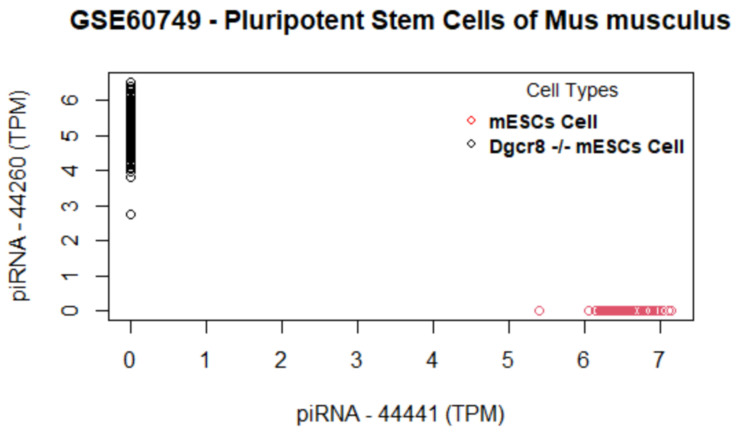
Data set GSE60749: Cells are clustered using the selected genes. The top gene (piRNA 44441) alone can perfectly differentiate two cell groups as there is no overlap of cell clusters.

**Figure 8 biology-11-01495-f008:**
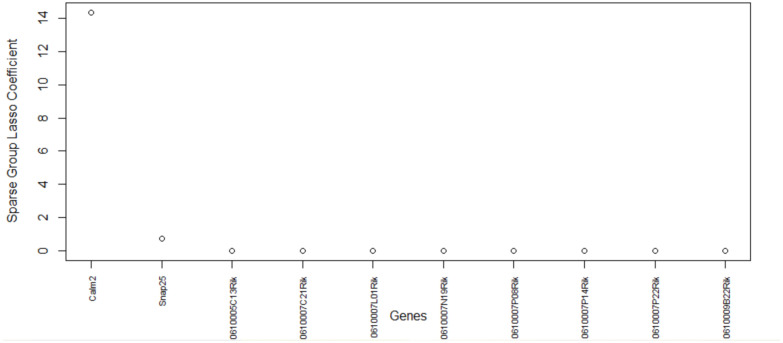
Genes versus coefficients plot for the data set GSE71585. Calm2 and Spap25 are the two most important genes.

**Figure 9 biology-11-01495-f009:**
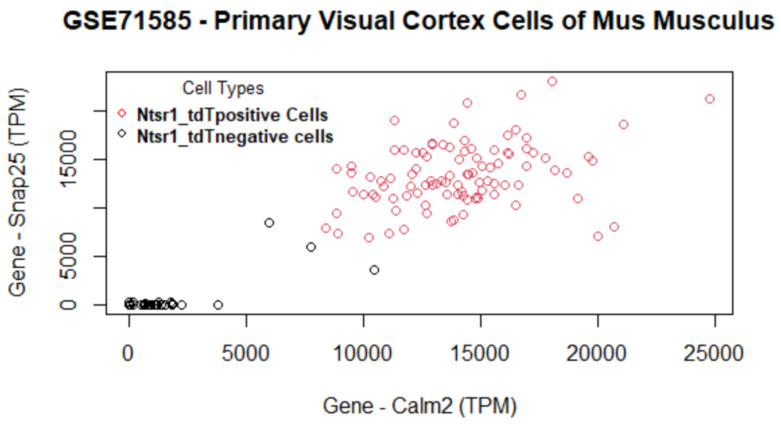
Data set GSE71585: Cells are clustered using the selected set of genes. The two most important genes (calm2 and Snap25) can differentiate two cell groups very well, as there is minimal overlap of cell clusters.

**Figure 10 biology-11-01495-f010:**
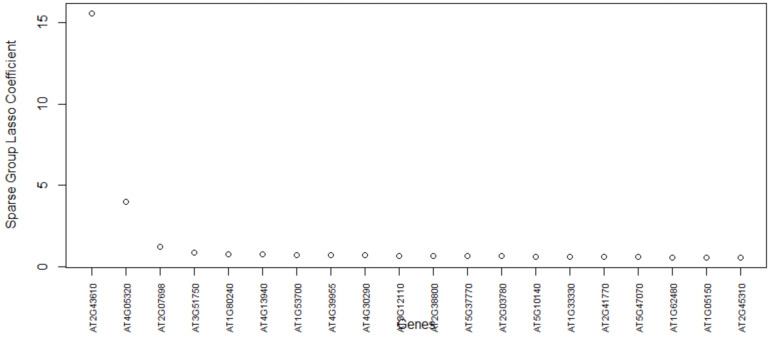
Genes versus coefficients plot for the data set GSE123818. Here, AT2G43610 and AT4G05320 are the two most important genes.

**Figure 11 biology-11-01495-f011:**
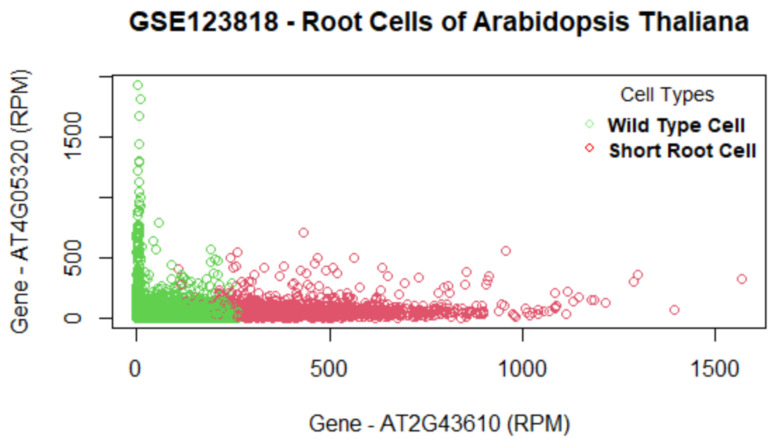
Data set GSE123818: Cells are clustered with the final set of selected genes. The top two important genes (AT2G43610 and AT4G05320) can differentiate the two cell groups with some overlap.

**Figure 12 biology-11-01495-f012:**
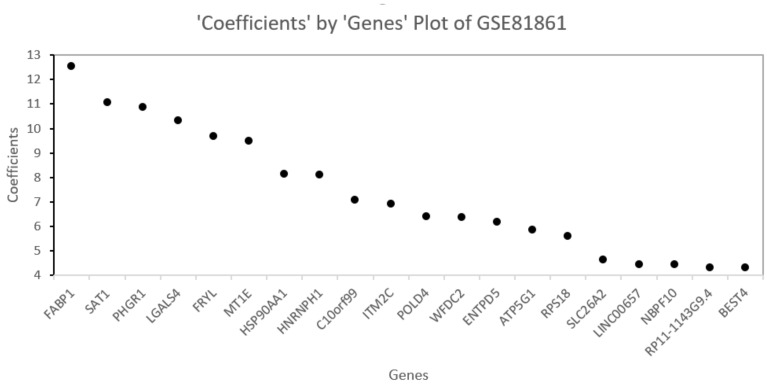
Genes versus coefficients plot for the data set GSE81861. FABP1 and SAT1 are the two most important genes.

**Figure 13 biology-11-01495-f013:**
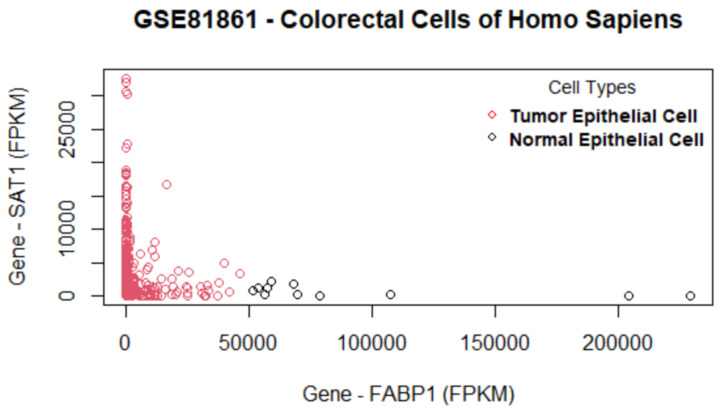
Data set GSE81861: Cells are clustered with the final set of selected genes. The two most important genes (FABP1 and SAT1) can differentiate two cell groups very well as there is some overlap of cell clusters.

**Figure 14 biology-11-01495-f014:**
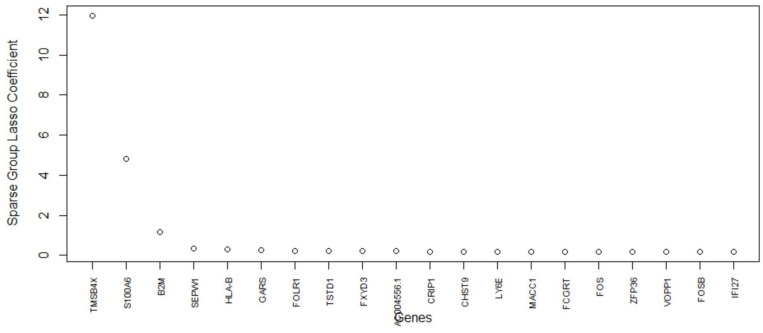
GSE157997 data: cell clustered (*K*-Means) with the final set of selected genes. The top two important genes are TMSB4X and S100A6. There is some overlap of the cell clusters from healthy donor and IPF patient.

**Figure 15 biology-11-01495-f015:**
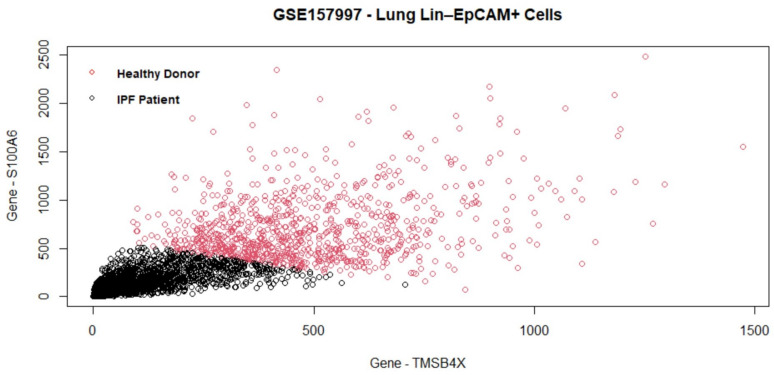
GSE157997 data: cell clustered (*K*-Means) with the final set of selected genes. The top two important genes are TMSB4X and S100A6. There is some overlap of the cell clusters from healthy donor and IPF patient.

**Table 1 biology-11-01495-t001:** Experimental data sets.

Data Set (Species)	Genes	Cells	Organism	Technology	Source
GSE60749 (Mouse)	22,443	183:84	Mus Musculus	Illumina HiSeq	conquer
GSE71585 (Mouse)	24,057	79:57	Mus Musculus	Fluidigm BioMark	conquer
GSE123818 (Plant)	27,629	1099:1099	Arabidopsis Thaliana	Illumina NextSeq	GEO
GSE81861 (Human)	57,241	272:160	Homo Sapiens	Fluidigm based scRNA-seq protocol	GEO
GSE157997 (Human)	33,694	1504:1504	Homo Sapiens	Illumina NextSeq 500	GEO

**Table 2 biology-11-01495-t002:** Performance comparison of the methods (Metric = Average CV-AUC). Note that the R packages for SGL, group lasso, and big lasso methods internally calculate models for a vector of λ values and select the λ (lambdamax), which minimizes prediction error. For ridge, lasso, and elastic net, the hyper-parameter λ is selected as lambda.1se. lambda.1se is the largest value of λ for which error is within 1 standard error of the cross-validated errors for the best model [45].

Method	R Package	60749	71585	123818	81861
Sparse Group Lasso (α=0.95, λ=lambdamax)	SGL	1	0.98	0.92	0.83
Group Lasso (λ=lambdamax)	grplasso	1	0.98	0.99	0.87
Drop Lasso (λ=0.001)	droplasso	0.99	0.94	0.97	0.87
Big Lasso (λ=lambdamax)	biglasso	1	1	0.95	0.80
Lasso (α=1,λ=lambda.1se)	glmnet	1	0.96	0.94	0.85
Elastic net (α=0.5,λ=lambda.1se)	glmnet	1	0.63	0.93	0.86
Ridge (α=0,λ=lambda.1se)	glmnet	0.99	0.84	0.90	0.71

**Table 3 biology-11-01495-t003:** Performance comparison for the methods (Metric = Average computation time in seconds).

Method	R Package	GSE60749	GSE71585	GSE123818	GSE81861
Sparse Group Lasso	SGL	6.53	1.73	5.66	2.97
Group Lasso	grplasso	1.12	2.51	3.78	29.66
Drop Lasso	droplasso	13.57	7.18	3.36	59.33
Big Lasso	biglasso	3.11	4.77	20.30	7.23
Lasso	glmnet	3.18	2.76	48.54	13.39
Elastic net	glmnet	3.57	2.66	51.51	13.59
Ridge	glmnet	58.07	26.71	17.58	3.77

**Table 4 biology-11-01495-t004:** Variance of CV-AUC.

Method	R Package	GSE60749	GSE71585	GSE123818	GSE81861
Sparse Group Lasso	SGL	0	0.0018	0.0012	0.0018
Group Lasso	grplasso	0	0.0018	0.0000	0.0017
Drop Lasso	droplasso	0.0004	0.0047	0.0003	0.0006
Big Lasso	biglasso	0	0	0.0061	0.0061
Lasso	glmnet	0	0.0029	0.0003	0.0017
Elastic Net	glmnet	0	0.0406	0.0007	0.0021
Ridge	glmnet	0.0004	0.0042	0.0003	0.0089

**Table 5 biology-11-01495-t005:** Comparison of performance (AUC) between SGL with all genes and SGL using the proposed method, which reduces the number of genes by the union of the selected genes from ridge, lasso, elastic net, and droplasso. The proposed method is consistently improving over SGL.

Data Set	All Genes	Gene Pool	SGL	Proposed Method
GSE60749	22,443	5965	1	1
GSE71585	24,057	5448	0.98	1
GSE123818	27,629	10,857	0.83	0.85
GSE81861	57,241	5823	0.92	0.94
GSE157997	33,694	15,386	0.9	0.9

**Table 6 biology-11-01495-t006:** Set of selected genes for GSE60749.

Gene	Function	Source
44441	piRNA. Function unknown.	ENA
44260	piRNA. Function unknown.	NCBI
44454	piRNA. Function unknown.	RNA Central
44446	Predicted gene. Function unknown.	RGD
44450	Predicted gene. Function unknown.	RGD
44440	piRNA. Function unknown.	piRNAdb
Pbld	Predicted to enable identical protein binding activity and isomerase activity. Predicted to be involved in maintenance of gastrointestinal epithelium; negative regulation of SMAD protein signal transduction; and negative regulation of transforming growth factor beta receptor signaling pathway.	NCBI
Lifr	Predicted to enable several functions, including ciliary neurotrophic factor receptor binding activity; growth factor binding activity; and leukemia inhibitory factor receptor activity. This gene has also been discussed in nine pathways including ESC pluripotency pathways.	NCBI
Hist2h4	It encodes a replication-dependent histone that is a member of the histone H4 family (basic nuclear proteins responsible for the nucleosome structure of the chromosomal fiber in eukaryotes). This gene is found in the Type II interferon signaling (IFNG) pathway.	NCBI
AK203176	Predicted to enable GTP binding activity; double-stranded RNA binding activity; and ubiquitin protein ligase binding activity. Acts upstream of or within cellular response to interleukin-4.	NCBI

**Table 7 biology-11-01495-t007:** Final set of selected genes for GSE71585.

Gene	Function	Source
Calm2	This gene enables calcium-dependent protein binding activity. It is involved in the Alzheimer’s disease pathway and Glycogen metabolism pathway. It is also involved in several important processes including regulation of response to tumor cell. Human ortholog of this gene is implicated in long QT syndrome 15.	NCBI
Snap25	This gene enables syntaxin-1 binding activity. It is used to study attention deficit hyperactivity disorder; obesity; schizophrenia; and type 2 diabetes mellitus. Human ortholog of this gene is implicated in Down syndrome and congenital myasthenic syndrome 18. It is found in 10 different pathways.	NCBI
0610005C13Rik	This gene is expressed in several structures, including heart; intestine; liver; lung; and metanephros.	NCBI
0610007C21Rik	This gene is replaced with name Atraid. It is predicted to be involved in several processes, including negative regulation of osteoblast proliferation; positive regulation of bone mineralization; and positive regulation of osteoblast differentiation.	NCBI

**Table 8 biology-11-01495-t008:** The final set of selected genes from GSE123818.

Gene	Function	Source
AT2G43610	This gene is found in growth and devlopmental stages such as root development. It enables chitinase activity and protein binding.	TAIR
AT4G05320	One of five polyubiquitin genes in Arabidopsis thaliana. These genes encode the highly conserved 76-amino-acid protein ubiquitin that is covalently attached to substrate proteins targeting most for degradation. This gene enables mRNA binding, protein tag, and ubiquitin protein ligase binding. The mRNA is cell-to-cell mobile.	TAIR
AT2G07698	This gene is expressed in growth and developmental stages such as seed and seedling development. It enables ADP binding, ATP binding, poly(U) RNA binding, and zinc ion binding.	TAIR
AT3G51750	This gene is expressed during initial leaves-visible stages and flowering stages. The biological processes associated with this gene are cellular lipid metabolic process, response to inorganic substance, response to light stimulus, root development, and seed development.	TAIR

**Table 9 biology-11-01495-t009:** Final set of selected genes from GSE81861.

Gene	Function	Source
FABP1	This gene encodes the fatty acid binding protein found in liver. Biological pathways—13, such as Peroxisome proliferator-activated receptor (PPAR) signaling pathway.	NCBI
SAT1	The protein encoded by this gene is a rate-limiting enzyme in the catabolic pathway of polyamine metabolism. Biological pathways—13. It has HIV-1 interaction and KFSD.	NCBI
PHGR1	It is a protein coding gene with biased expression in colon and small intestine.	NCBI
LGALS4	The galectins are implicated in modulating cell–cell and cell–matrix interactions. The expression of this gene is restricted to the small intestine, colon, and rectum. It is underexpressed in colorectal cancer.	NCBI
FRYL	This gene is predicted to be involved in cell morphogenesis and neuron projection development. It is predicted to be active in the site of polarized growth.	NCBI
MT1E	Biological pathways—5, such as Zinc homeostasis and Copper homeostasis. HIV-1 Tat upregulates the interferon-responsive gene expression of Metallothionein, an effect that likely facilitates the expansion of HIV-1 infection.	NCBI
HSP90AA1	The protein encoded by this gene aids in the proper folding of specific target proteins. Biological pathways—115, such as programmed cell death and innate immune system. It has strong interactions with HIV-1 proteins.	NCBI
HNRNPH1	This gene may be associated with hereditary lymphedema type I. Biological pathways—12, such as mRNA processing. Knockdown of HNRNPH1 inhibits the early stages of HIV-1 replication in 293T cells.	NCBI

**Table 10 biology-11-01495-t010:** Final set of selected genes for GSE157997.

Gene	Function	Source
TMSB4X	This gene encodes an actin sequestering protein that plays a role in the regulation of actin polymerization. The protein is also involved in cell proliferation, migration, and differentiation. It is found in three pathways—notably, the VEGFA-VEGFR2 signaling pathway. This pathway is related to Angiogenesis. Angiogenesis is the formation of new blood vessels from pre-existing vasculature; is central to several physiological conditions, from embryogenesis to wound healing in adults; and is a hallmark of pathological conditions such as tumorigenesis. Tumorigenesis is the gain of malignant properties in normal cells, including primary dedifferentiation, fast proliferation, metastasis, evasion of apoptosis and immunosurveillance, dysregulated metabolism and epigenetics, etc., which have been generalized as the hallmarks of cancer.	NCBI
S100A6	Chromosomal rearrangements and altered expression of this gene have been implicated in melanoma. It is involved in the up-regulation of fibroblast proliferation.	NCBI
B2M	The encoded antimicrobial protein displays antibacterial activity in amniotic fluid. A mutation in this gene has been shown to result in hypercatabolic hypoproteinemia. Related to 25 pathways including disease, HIV, and immune system.	NCBI
SEPW1	Studies in mice show that selenoprotein is involved in muscle growth and differentiation, and in the protection of neurons from oxidative stress during neuronal development. This gene is related to two pathways.	NCBI
HLA-B	The protein encoded by this gene plays a central role in the immune system. This gene has HIV interaction and is linked to four pathways. This gene is particularly relevant to IPF because immunity is a pathogenesis of IPF [44].	NCBI

## Data Availability

All the data used are downloaded from public database and the links are provided in the paper. No new scRNA-seq data were generated or created.

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
