# Peer review of "A Novel Algorithm for Feature Selection Using Penalized Regression with Applications to Single-Cell RNA Sequencing Dataâ€"

_biology, 2022, doi:10.3390/biology11101495_

Round 1
Reviewer 1 Report
Authors did a good job to list all the penalized regression techniques for the analysis of single-cell RNA-sequencing (scRNA-seq) data. Besides, there are certain comments need to be addressed before going for any decision.
Major Comments
1. I think the high-dimensionality in scRNA-seq studies is a major challenge, as the platforms like drop-seq can generate data of thousands of transcripts up to millions of cells.
2. Authors tried to describe “large-p-small-n" problem but did not mention the consequences of “large-p-small-n" in estimation or data analysis.
3. In methods section, authors listed the available penalized regression techniques with greater detail, which are well known in literature. Authors shall try to provide more space to their proposed method instead focusing more on the techniques, which are in public domain.
4. Authors mentioned that they have proposed one method, which I could not find in the methods section. They have listed all the existing techniques for penalized regression and clustering. Apparently, the proposed method must have given more space in this article, but could not find the proposed ‘ensemble of penalized regression’ in the method section.
5. Authors may clarify why simulation was not considered for benchmarking the penalized regression methods and only 5 real datasets are considered.
6. If SGL is the best performed technique for feature selection, why other techniques ridge, lasso, elasticnet and droplasso (figure 1) are considered in the proposed algorithm for feature selection.
Minor Comments
1. Abstract: “…Sparse Group Lasso (SGL) outperforms the other 6 methods…” shall name all the six methods for better understanding of the readers.
2. Introduction: “…scRNA-seq protocols can be classified into two categories, full-length transcript sequencing approaches and 3’-end or 5’-end transcript sequencing...” more clarification is required regarding the two categories with suitable example. Also consider to cite few articles here [ref. PMID: 35885218; PMID: 34946896]
3. Add “,” in line 47 after “in this research”.
4. Line 57 is not clear, restructure it.
5. Grammatical and spell errors throughout the manuscript, consider a careful revision. For instance, “Mathematically speacking…”. Line 272, ‘snRNA-seq’.
6. Some of the tables and figures are not so informative, so may be kept in Supplementary to reduce space.
Reviewer 2 Report
In this paper, the authors proposed a comparison study for regularization regression. Overall, I support this paper. The following lists some comments. First about the regression. As for the classification, the authors need refer another similar work in a recent paper (DOI:10.1007/s10489-021-02877-3) and the references therein. Second is about the data. This paper focuses on scRNA-seq data. It is not clear how to impute the data as well as the influences of the sample size for the classification. Third is about the ensemble method. It is not clear how to integrate the selected features by these methods to generate a final results. This paper is also important for this paper. Minor comment is about the contributions need be clearly presented. Thanks.
Reviewer 3 Report
Annotating different cell types in single-cell RNA-Seq is challenging and requires prior knowledge to annotate cell clusters based on the genes/markers. The complexity and annotation become more challenging as the number of cell types and clusters increases.
Here, the manuscript by Puliparambil and colleagues use mouse, human and plant single-cell datasets and developed a pipeline/ensemble for classification of 2-cell clusters (binary classification) and evaluated/compared the power of different methods. While the manuscript is well written and explained, the real single-cell data analyses with many cell clusters is more challenging. The authors indicated clearly (line-326), that they only use binary classification but not multi-classification. However, the study remains low quality by just using binary classification, while in reality it requires multi-classification. A binary classification may not be enough to make comparisons and derive conclusions. The authors should use multiclassification for their analyses.
Additionally, it is not clear the biological relevance of the genes discussed and the cell types used. As there is no batch correction, the genes selected could be differentially expressed genes rather than markers. The top marker genes/piRNAs could be because of treatment/knock-out etc, but not marker genes. At least for the current reviewer, it is not clear.
The authors should discuss if there is any overlapping or differences between the selected genes and the two classes of cells (based on the original studies).
The genes discussed in the Discussion may indicate differently expressed genes. For instance, the top genes that are piRNAs is most likely because of knock-out DGCR8 (which play role in small RNA biogenesis).
It is not clear, if the binary classification takes into account the biological effect versus the cell types.
The authors should share the source code for the reproducibility of the analyses. This may be done on GitHub.
Minor comments:
The numbers in Tables and in the text do not match and must be corrected. e.g., table-1, GSE60749 (224444) (in text line-253, 22443), same for line-258.
After line-221; the equation has been repeated.
there are many typos that must be corrected.
Round 2
Reviewer 3 Report
The authors addressed all of my questions. I do not have any further comments.